# Rosemary as a Potential Source of Natural Antioxidants and Anticancer Agents: A Molecular Docking Study

**DOI:** 10.3390/plants13010089

**Published:** 2023-12-27

**Authors:** Haytham Bouammali, Linda Zraibi, Imane Ziani, Mohammed Merzouki, Lamiae Bourassi, Elmehdi Fraj, Allal Challioui, Khalil Azzaoui, Rachid Sabbahi, Belkheir Hammouti, Shehdeh Jodeh, Maryam Hassiba, Rachid Touzani

**Affiliations:** 1Laboratory of Applied Chemistry Environment (LCAE), Faculty of Science Oujda, University Mohammed First, Oujda 60000, Morocco; hermione_1995@hotmail.fr (I.Z.); moh.merzouki@gmail.com (M.M.); lamizenaga@gmail.com (L.B.); elmehdipereira@gmail.com (E.F.); allal.challioui@gmail.com (A.C.); r.touzani@ump.ac.ma (R.T.); 2Water, Environment and Sustainable Development Laboratory (LEEDD), Faculty of Science Oujda, University Mohammed First, Oujda 60000, Morocco; zraibilinda@gmail.com; 3Laboratory of Engineering, Electrochemistry Modeling and Environment, Faculty of Sciences, Sidi Mohamed Ben Abdellah University, Fez 30000, Morocco; k.azzaoui@yahoo.com; 4Euro-Mediterranean University of Fes (UEMF), Fez 30070, Morocco; r.sabbahi@gmail.com (R.S.); hammoutib@gmail.com (B.H.); 5Higher School of Technology, Ibn Zohr University, Quartier 25 Mars, P.O. Box 3007, Laayoune 70000, Morocco; 6Laboratory of Industrial Engineering, Energy and the Environment (LI3E), SupMTI, Rabat 10000, Morocco; 7Department of Chemistry, An-Najah National University, Nablus P.O. Box 7, Palestine; sjodeh@najah.edu; 8College of Medicine, QU Health, Qatar University, Doha 2713, Qatar

**Keywords:** carnosic acid, cell signaling, molecular docking, oxidative stress, proliferation, rosmarinic acid, *Rosmarinus officinalis* L., survival, tumor

## Abstract

*Rosmarinus officinalis* L. compounds, especially its main polyphenolic compounds, *carnosic acid* (CA) and *rosmarinic acid (RA*), influence various facets of cancer biology, making them valuable assets in the ongoing fight against cancer. These two secondary metabolites exhibit formidable antioxidant properties that are a pivotal contributor against the development of cancer. Their antitumor effect has been related to diverse mechanisms. In the case of CA, it has the capacity to induce cell death of cancer cells through the rise in ROS levels within the cells, the inhibition of protein kinase AKT, the activation of autophagy-related genes (ATG) and the disrupt mitochondrial membrane potential. Regarding RA, its antitumor actions encompass apoptosis induction through caspase activation, the inhibition of cell proliferation by interrupting cell cycle progression and epigenetic regulation, antioxidative stress-induced DNA damage, and interference with angiogenesis to curtail tumor growth. To understand the molecular interaction between *rosemary* compounds (CA and RA) and a protein that is involved in cancer and inflammation, S100A8, we have performed a series of molecular docking analyses using the available three-dimensional structures (PDBID: 1IRJ, 1MR8, and 4GGF). The ligands showed different binding intensities in the active sites with the protein target molecules, except for CA with the 1MR8 protein.

## 1. Introduction

Aromatic plants have emerged as promising contenders in the realm of cancer treatment. They offer bioactive compounds with potent medicinal properties that can stop tumors from growing and induce cancer cell death. Notable studies have revealed that these secondary metabolites exhibit formidable antioxidant properties which are a pivotal contributor against the development of cancer [1]. Additionally, compounds derived from this natural resource have demonstrated their ability to thwart the activity of cancer stem cells, which are pivotal actors in tumor progression and metastasis. This compelling body of evidence underscores the significant promise held by aromatic plants as a wellspring of natural compounds for advancing cancer treatment [2].

In the vast realm of bioactive compounds, polyphenols are a prominent class of secondary plant metabolites that demand attention due to their multifaceted bioactive properties [3]. Despite their structural diversity, polyphenols play pivotal roles in various biological processes within plants, substantially contribute to the sensory and nutritional attributes of plant-based foods, and have potential applications in a wide array of practical contexts (formulation of traditional medicine, pharmaceutical industry, food processing and preservation) [4,5].

Rosemary, scientifically known as *Rosmarinus officinalis* L., as a cornerstone in both economic and social realms, plays a vital role in the countries where it thrives. Its significance is multifaceted, contributing substantially to local economies and societal practices [3]. With an estimated annual production ranging from 150 to 200 tons, rosemary has emerged as a key player in agriculture and industry. Notably, Tunisia leads production with 80 tons, followed by Morocco with 40 tons, Spain with 28 tons, and France contributing 5 to 10 tons of rosemary oil [4,6]. The cultivation of rosemary not only supports farmers but also fuels the agribusiness sector, providing employment opportunities in rural areas [7]. The extraction of its essential oil, widely utilized in the fragrance and flavoring industries, contributes to international trade, fostering economic growth and generating export revenue. Beyond its economic impact, rosemary is deeply interwoven into the social fabric of cultures [8]. Its aromatic and flavorful properties make it a staple in culinary traditions, enhancing the taste of diverse dishes and contributing to a sense of cultural identity. Additionally, rosemary holds a rich history in traditional medicine, where its potential health benefits, including hepatoprotective, antifungal, insecticidal, antioxidant and antibacterial effects [9], contribute to traditional healing practices. The cultural significance of rosemary extends to rituals, ceremonies, and festivals, adding to the cultural richness of communities [10]. Apart from the wide spread’s use of its essential oils in various applications, numerous scientific studies have explored the possible benefits of rosemary polyphenols [8] and their individual components, particularly those with antioxidant effects, for treating inflammation, liver damage, and cancer. Nowadays, in the European Union, rosemary extracts are added to food products and beverages at levels of up to 400 mg/kg, assigning them the label E392 [11].

This review offers a comprehensive exploration of the pivotal role played by natural products derived from *rosemary extracts* (Res) in the realms of oxidative damage and cancer research. Its primary focus centers on the potent anticancer potential of *rosmarinic* and *carnosic acids* that are the most compelling compounds sourced from rosemary. Furthermore, the review employs molecular docking techniques to unveil the intricate interactions between these phenolic powerhouses and a critical drug target, S100A8, offering insights into their therapeutic promise. Notably, this review provides the depth analyses of three less-explored cancer types: oral, kidney, and pancreatic cancer. It delves into the comparative effects of *rosmarinic acid* and *carnosic acid* on these selected cancers, adding the grasp of their potential in combating these challenging malignancies. In essence, aromatic plants like rosemary, with their rich reservoir of bioactive compounds, hold significant promise in the ongoing battle against cancer.

## 2. Potential Antioxidant and Anticancer Effects of Res

### 2.1. Polyphenols: Diverse Roles in Plants and Humans

Polyphenols are secondary metabolites that are widely found in different plants and play a pivotal role in both plant biology and human health [12]. These compounds exhibit a wide range of structural diversity and encompasses phenolic acids, phenolic alcohols, and other molecules with multiple hydroxyl groups on aromatic rings [13]. In our diet, polyphenols are commonly found in foods like fruits, vegetables, cereals, legumes, tea, coffee, and chocolate [14]. However, the quantity and quality of polyphenols in these foods depend on factors such as plant genetics, growing conditions, soil composition, harvest maturity, and post-harvest handling [15]. Polyphenols have antioxidant effects and can scavenge free radicals due to their chemical structure. However, the antioxidant activity depends on some parameters such as the type of compound, the level *of methoxylation*, and the number of hydroxyl groups [16]. Indeed, this property is responsible for the protection against several diseases (cancer, Alzheimer’s disease, cardiovascular disorders) Richheimer et al. [17]. Recent research by Birtić et al. [18] sheds light on the potential of natural products, including polyphenols, to interact with microtubule affinity regulatory kinase (MARK 4) and promising candidate for cancer treatment. Interestingly, polyphenolic antioxidants can exhibit a dual role as prooxidants. This can trigger apoptosis in cancer cells and damage biological macromolecules, mainly in systems that contain redox-active metals such as copper [19] (Figure 1).

In line with the exploration of polyphenols, Fadili et al. [21] conducted a comprehensive analysis to quantify the polyphenol content in two distinct species of rosemary. Their investigation unveiled the presence of a rich array of secondary metabolites within both plant species, with a particular focus on the abundance of *polyphenols*. These *polyphenols* exhibited variable concentrations, ranging from 0.02 to 0.185 g, signifying the remarkable diversity in secondary metabolite content. Furthermore, the researchers endeavored to assess the antioxidant potential of these compounds using the DPPH method. The results were short and impressive, particularly for the ethyl acetate fractions, which displayed robust DPPH radical-scavenging abilities. In fact, the inhibitory concentrations responsible for 50% of the antiradical activity (IC50) were measured at an astonishing 103.86 ± 3.5 μg/mL for *Rosmarinus officinalis* L. To determine the precise polyphenol content within the plant extracts, UV/visible spectrophotometry was employed. The outcomes of this meticulous analysis are visually presented in Figure 2, showcasing the rich polyphenolic landscape of these rosemary species.

### 2.2. Phenolic Powerhouses in Rosemary: Carnosic and Rosmarinic Acids

A great number of compounds were identified in a solid liquid extract from rosemary, using, most often, HPLC–MS and GC-MS as methods of analysis. These secondary metabolites belong mainly to the classes of phenolic diterpenes (*carnosic acid*, *carnosol, rosmadial*, *rosmanol*), *phenolic acids* (*rosmarinic acid*, *caffeic acid*), *flavonoids* (*genkwanin*, *homoplantaginin*, *scutellarein and cirsimaritin*), and *triterpenes* (*ursolic acid*, *oleanolic acid*) (Figure 3) [22]. Their levels vary significantly based on the parts of the plant (leaves, stems, sepals, petals, seeds, roots), seasonal variation, and the extraction method [23].

The principal bioactive components of the rosemary extracts are *carnosic acid* (CA), *rosmarinic acid* (RA), *and carnosol* (CAR) [5]. CA and RA, prominent *phenolic* compounds in rosemary leaves, boast unique attributes that contribute to their significant roles in health and well-being. CA, characterized by its abietane-like structure fused with a *catechol* group and *O-phenolic hydroxyl* groups at positions C11 and C12 [24], distinguishes itself by accumulating preferentially in rosemary’s glandular trichomes. Its lipophilic nature empowers CA to excel the antioxidant activity, effectively countering singlet oxygen, hydroxyl radicals, and lipid peroxyl radicals. Beyond its antioxidative prowess, CA exhibits a diverse repertoire of benefits, including anti-inflammatory, antiproliferative, antitumorigenic, and neuroprotective properties [25]. Recent research underscores CA’s substantial contribution, along with its primary oxidation product, carnosol (CAR), comprising around 90% of rosemary extract’s antioxidant capabilities [26]. This remarkable antioxidant response owes much to CA’s abundant presence relative to other *phenolic* compounds. Intriguingly, CA’s superior protection against lipid peroxidation, as evidenced in accelerated aging experiments, surpasses that of *α-tocopherol*, while its rapid consumption leads to the absence of accumulated conjugated *hydroperoxides* of polyunsaturated fatty acids [27].

Rosemarinic acid (RA) has a chemical structure termed 3,4-dihydroxyphenyllactic acid and displays versatility through a spectrum of health-enhancing qualities. Widely distributed across plant families, including Boraginaceae and Lamiaceae [28], RA shines with its antiviral, antibacterial, anti-inflammatory, and antioxidant effects [29]. What is intriguing is RA’s potential for synthesis in undifferentiated plant cell cultures, often yielding higher levels than the original plant source. Its documented ability to alleviate inflammation by reducing Cyclooxygenase 2 enzyme (COX 2) expression and prostaglandin levels further highlights its significance. Nevertheless, when assessed for antioxidant potency through DPPH tests, CA, with its two phenolic hydroxyl groups, surpasses RA, which boasts four such groups (Table 1) [30]. Together, CA and RA, as major polyphenolic constituents in rosemary extract, exhibit formidable antioxidant properties. Research has explored how these substances protect against oxidative stress and glycation in diabetic rats, revealing a decrease in oxidative stress markers and advanced glycation end products. This signifies their ability to combat oxidative damage and glycation, offering potential benefits in mitigating oxidative stress in diabetes [31].

## 3. Charting the Antioxidant Odyssey: Carnosic Acid and Rosmarinic Acid in the Fight against Different Type of Cancer

*Rosemary*, along with its notable derivatives like CA and RA, has firmly established itself as a formidable force in the ongoing battle against cancer [39]. These compounds showcase a multifaceted array of anticancer activities, with their effectiveness intricately woven into various mechanisms that directly impact the complex landscape of cancer biology [40]. Of particular significance is the spotlight on their potent antioxidant effects and features that has garnered substantial attention within the realm of cancer research. This attribute equips them with a remarkable ability to combat the insidious menace of oxidative stress, a pivotal contributor to the development of cancer.

Their prowess lies in their adeptness at neutralizing free radicals, thus acting as staunch defenders of the integrity of DNA, proteins and lipids [41]. By shielding these vital cellular components from the perils of oxidative damage, these compounds have an essential function in thwarting the initiation and progression of tumors. The significance of this antioxidant action becomes evident when considering its role in decreasing the risk of tumor initiation and restraining its relentless advance. However, the narrative takes an intriguing twist when delving into the nuanced duality of these antioxidants [42].

While they predominantly stand as guardians against oxidative harm, it is crucial to acknowledge their ability to transform into potent warriors, particularly under specific conditions. This transformation results in the release of reactive oxygen species (ROS) endowed with cytotoxic effects. This delicate balance, where antioxidants provide fortification against oxidative damage while selectively inducing ROS-mediated cytotoxicity, forms the crux of the overall anticancer potential embodied by rosemary compounds [33].

Unraveling this intricate relationship between antioxidants and their dual roles, one as protective shield against oxidative stress and the other as orchestrators of targeted cytotoxicity, unveils invaluable insights into their therapeutic promise in cancer management. However, it is essential to underline that the precise mechanisms through which CA and RA exert their effects, as well as their efficacy, may exhibit variations across different types of cancer. In the forthcoming sections, we will embark on a comprehensive exploration of the distinct roles played by CA and RA in combating various forms of cancer. This journey will illuminate the intricate interplay between their antioxidant prowess and their stature as indispensable assets in the ongoing war against cancer.

Intriguingly, carnosic acid has demonstrated its ability to induce cell death by activating phenomena involving protein kinase B (AKT) inhibition. This dual mechanism enhances the efficacy of CA as an antitumor agent, positioning it as herbal medicine that can both prevent and treat some cancers [43]. Furthermore, CA’s potential link between autophagy and apoptosis has emerged as a pivotal aspect leading to cell death in hepatocellular carcinoma cells (HepG2). CA’s stimulation of autophagic vacuoles through the activation of autophagy-related genes (ATG), possibly ATG-12, triggers caspase activation and inhibits the antiapoptotic protein Bcl-2, both pivotal in apoptosis. Additionally, CA has the capacity to disrupt mitochondrial membrane potential, causing the release of cytochrome c (cyt c), nuclear cleavage of peroxisome proliferation-activated receptor (PARP), and activation of pro-apoptotic proteins Bax/Bcl-2, all contributing to apoptosis. The down-regulation of AKT and possibly other different molecular pathways is involved in these complex processes that, together, induce HepG2 cell death (Figure 4).

In parallel, *rosmarinic acid* (RA) has also exhibited a diverse range of anticancer mechanisms. RA’s actions encompass apoptosis induction through caspase activation, the inhibition of cell proliferation by interrupting cell cycle progression, and its anti-inflammatory properties, which suppress inflammation within the tumor microenvironment. RA’s role as a potent antioxidant guards against oxidative stress-induced DNA damage, and its interference with angiogenesis holds potential in curtailing tumor growth [44]. In addition to that, RA may engage in epigenetic regulation, altering gene expression patterns that influence cancer progression. This versatile compound can also modulate autophagy, sometimes leading to autophagic cell death. Collectively, RA’s multifaceted actions position it as a promising candidate for cancer therapy, albeit with precise mechanisms varying depending on cancer type and cellular context. Ongoing research aims to further unravel the intricate molecular pathways underlying RA’s anticancer effects. According to the selected studies, rosemary and its main compounds influence various facets of cancer biology, which makes them valuable assets in the ongoing fight against cancer.

### 3.1. Oral and Ovarian Cancers

Oral cancer presents an alarming challenge in the realm of oncology, with 5-year survival rates hovering around 50%, despite treatment modalities such as radiotherapy and chemotherapy. The intricate pathogenesis of oral cancer involves a multifaceted interplay of genetic alterations within tissues exposed to carcinogens. Addressing this complex issue, Lindgren et al. [45] conducted a study aimed at unraveling the potential anticancer properties of rosemary extract (RE) specifically within the context of oral squamous cell Carcinoma (OSCC). Their comprehensive investigation encompassed diverse facets, including antioxidant activity, cytotoxicity, autophagy induction, and the expression of apoptotic markers, notably caspase-3, in OSCC. This study revealed valuable knowledge about how RE can be useful for treating OSCC. First and foremost, the evaluation of antioxidant activity demonstrated that RE has the capacity to counteract oxidative stress, a significant driver of cancer progression. Furthermore, RE displayed notable cytotoxicity against OSCC cells, suggesting its potential as a therapeutic agent. Intriguingly, RE’s induction of autophagy in OSCC cells showcases its diverse anticancer mechanisms. Additionally, Khella et al. [33] conducted another study that highlighted RE’s significant antiproliferation activity on both A2780 and A2780CP70 ovarian cancer cell lines, with IC50 values estimated at 1/1000 dilution for A2780 cells and 1/400 dilution for A2780CP70 cells. This underscores the broad spectrum of RE’s antiproliferative effects across different cancer types.

However, a more nuanced understanding emerges when comparing the individual components of RE, like *carnosic acid* and *rosmarinic acid*. CA and RA exhibited distinct IC50 values in relation to RE, underscoring the potency of RE as an antiproliferative agent against OSCC. Notably, CA demonstrated an IC50 of 7.5 μg/mL for A2780 cells, whereas RA exhibited a substantially higher IC50, exceeding 50 μg/mL. This disparity suggests that CA contributes significantly to RE’s antiproliferative effect. A similar trend was observed in A2780CP70 cells, with CA’s IC50 exceeding 20 μg/mL, while RA displayed no significant antiproliferation activity at 50 μg/mL. These results emphasize the synergistic action of RE’s constituents, with CA playing a pivotal role.

In conclusion, those studies illuminate the potential of RE as a valuable asset in managing OSCC, shedding light on its multifaceted anticancer mechanisms encompassing antioxidant activity, cytotoxicity, autophagy induction, and apoptotic regulation. Furthermore, the comparative analysis of CA and RA highlights their distinct contributions to RE’s antiproliferative effects, underscoring the importance of their combined action. By revealing these findings, new avenues for research and creative approaches are opened in the continuous will fight against oral cancer.

### 3.2. Kidney Cancer

Kidney cancer, a disease where cells in one or both kidneys grow out of control, primarily manifests as renal cell carcinoma in adults. The following studies shed light on the potential implications of CA and RA in the context of kidney cancer.

Mina et al. delved into the effects of CA on human renal carcinoma Caki cells, revealing intriguing outcomes [46]. Their investigation unveiled that CA led to a rise in ROS levels within the cells. Notably, the authors found that ROS inhibitors, specifically N-acetyl cysteine (NAC) and glutathione ethyl ester (GEE), were able to mitigate the activity of ATF4 (Activating transcription factor 4), a protein central to stress response regulation; and CHOP (C/EBP homologous protein), a protein responsible for inducing cell death, in CA-treated cells. This observation hinted at CA’s role in inducing cell death by elevating ROS levels and triggering endoplasmic reticulum (ER) stress in renal carcinoma cells. Subsequently, the research was extended to explore how CA influenced the sensitivity of cancer cells to TRAIL (TNF-related apoptosis-inducing ligand), a protein known to initiate apoptosis in cancer cells. The study findings indicated that CA meaningfully augmented the expression of DR5, Bim, and PUMA, proteins that promote TRAIL-induced apoptosis. Concurrently, CA led to a reduction in the expression of c-FLIP and Bcl-2, proteins that typically inhibit TRAIL-induced apoptosis. This dual action of CA appeared to enhance the effectiveness of TRAIL in targeting cancer cells and potentially overcoming TRAIL resistance in these cells [47].

In another study carried out by Chou et al. [37] focusing on kidney cell lines, the effects of RA on HK-2 and 786-O cells were investigated. These cells were subjected to varying concentrations of cisplatin (CDDP) in combination with RA, and a series of assays were conducted to assess cell viability, migration, and cell cycle analysis. The results yielded significant insights into the potential therapeutic interventions for kidney-related diseases. RA and CDDP exhibited inhibitory effects on the growth of HK-2 and 786-O cells in a dose-dependent manner, as evidenced by the MTT test measuring cell viability. Furthermore, the study assessed their effect on cell movement using transmigration and wound closure tests. These investigations revealed that RA and CDDP significantly curtailed cell migration compared to the control group, again in a dose-dependent manner. Collectively, these findings indicate that RA and CDDP hold promise as potential therapeutic approaches to target the growth and migration of kidney cells, suggesting their relevance in managing kidney-related diseases.

According to these selected studies, rosemary-derived compounds, including CA and RA, exhibit significant potential in addressing the complexities of kidney cancer, both in terms of inducing cell death and inhibiting cell migration. These findings will enable further research and the development of innovative strategies to combat kidney cancer.

### 3.3. Pancreatic Cancer

Pancreatic cancer, a highly lethal malignancy with a rising incidence and a low survival rate [48], has also been the focus of studies exploring the anticancer effects of rosemary Reid et al. [49] conducted investigations into the potential of supercritical rosemary extract (ER) on both pancreatic and colon cancer cells. Their findings revealed that colon cancer cells exhibited greater sensitivity to the extracts compared to pancreatic cancer cells, leading to a decrease in cell viability. Additionally, this study observed that the extracts influenced the expression of GCNT3, a metabolic gene, and a microRNA known as miR-15b, both of which play roles in the epigenetic regulation of cancer. Intriguingly, miR-15b levels were found to decrease in the blood of mice following rosemary treatment.

*Carnosic acid*, and RA, both garnered attention for their potential effects on pancreatic cancer. CA has demonstrated its aptitude to induce apoptosis and impede the proliferation of pancreatic tumor cells. Its actions are directed towards the Nrf2/ARE signaling pathway and it is a key player in cell growth and survival. On the other hand, RA has been recognized for its antioxidant and anti-inflammatory properties, which could play a role in its potential anticancer effect.

To explore the anticancer effects of RA in pancreatic cancer, both in vitro and in vivo experiments were performed using pancreatic cell lines and nude mice with xenografts. In vitro experiments, including the MTT assay, showed a concentration and time-related decrease in cell survival for the Panc-1 and SW1990 cell lines, with more than 50% reduction observed at 100 µM [50]. Moreover, these experiments revealed a significant reduction in cell growth, invasion, and migration. RA treatment also triggered apoptosis in the pancreatic cancer cell line and lowered the expression of epithelial-mesenchymal transition markers such as vimentin and N-cadherin.

Notably, RA treatment was found to up-regulate miR-506 levels in a concentration-dependent manner in both cell lines. Luciferase reporter assays confirmed that miR-506 targeted the 3′ untranslated region of MMP-2 and MMP-16 genes. Subsequently, RA treatment suppressed the overexpression of MMP-2 and MMP-16, resulting in the inhibition of cell invasion and migration. These promising in vitro findings were further corroborated by in vivo studies in a xenograft nude mice model, where RA demonstrated its potential to suppress pancreatic tumor growth and related mechanisms. Altogether, these findings reveal how RA works as a potential antitumor compound for pancreatic cancer, regulated by the miR-506/MMP2/16 pathway.

According to the literature, rosemary, particularly via its compounds CA and RA, holds considerable promise in addressing the challenges posed by pancreatic cancer. This presents an approach that warrants further exploration for potential therapeutic strategies.

### 3.4. Colon Cancer

Colorectal cancer (CRC) ranks as the third most prevalent neoplastic malignancy in Asia, constituting 9.7% of all cancer cases. Predominantly affecting individuals aged over 50, developed nations show higher incidence rates due to lifestyle changes [50]. Despite lower prevalence, Asia bears the highest number of CRC cases [51]. In 2013, CRC stood as the fourth leading cause of global cancer-related deaths, with 19.3 million new cases and 9.9 million deaths in 2020 [52]. Treatment options include surgery, radiotherapy, immunotherapy, targeted therapy, and chemotherapy, but the cure or survival rate has not significantly improved. Risk factors encompass age, Western diets, and genetic mutations. CRC typically initiates as mucous membrane polyps, with genetic mutations accumulating through mechanisms like chromosomal instability [50].

Rosemary and its compounds, *carnosic acid* (CA) and *rosmarinic acid* (RA), have been studied for their potential effects on colon cancer. CA inhibits HCT116 and SW480 colon cancer cell proliferation via the Nrf2/ARE signaling pathway, promoting antioxidant defense. RA, another rosemary compound, induces apoptosis and inhibits cell proliferation in HCT116 and SW480 colon cancer cell lines, targeting the Nrf2/ARE signaling pathway [39]. Pérez-Sánche et al. [39]’s study demonstrates rosemary extract’s ability to inhibit tumor growth in vivo, suppressing proliferation, migration, and colony formation of colon cancer cells. This occurs through increased intracellular reactive oxygen species (ROS), leading to necrotic cell death. The inhibition of colony formation ranged from 76.9% to 92.3% in HGUE-C-1 cells, 76.9% to 87.1% in SW480 cells, and 52.3% to 84.5% in HT-29 cells, depending on the concentration of rosemary extract [39]. *Carnosic acid* (CA) exhibits inhibitory effects on colorectal cancer cell growth and migration, inducing apoptosis in liver cancer cells through the ROS-mediated mitochondrial pathway [40]. CA-loaded polymeric nanoparticles, specifically bovine serum albumin (BSA) nanoparticles, enhance CA’s antitumor activity against colorectal cancer cells. These nanoparticles demonstrate heightened efficacy, with lower IC50 values (2.60 μg/mL and 6.02 μg/mL) against Caco-2 and MCF-7 cells, respectively, compared to free CA (8.29 μg/mL and 27.43 mg/mL). DNA content analysis indicates cell growth arrest at the G2/M phase, and the apoptosis assay shows higher total apoptosis in cells treated with CA-BSA nanoparticles [40].

Han et al. [41]’s in vitro study on *rosmarinic acid* (RA) in metastatic colorectal cancer cells reveals time- and concentration-dependent inhibition of cell proliferation. RA induces cell cycle arrest in the G0/G1 phase, regulates epithelial–mesenchymal transition (EMT), and inhibits migration, invasion, and adhesion of CRC cells. The study emphasizes RA’s impact on reducing matrix metalloproteinases (MMP-2 and MMP-9) and downregulating antiapoptotic genes (BCL-2 and COX-2) [41]. Overall, these findings highlight RA’s potential as a multifaceted therapeutic agent against metastatic colorectal cancer. In the study by Han et al. [41], RA demonstrates a time- and concentration-dependent inhibition of colorectal cancer cell proliferation, inducing cell cycle arrest in the G0/G1 phase. RA also regulates epithelial–mesenchymal transition (EMT), inhibits migration, invasion, and adhesion of CRC cells, and reduces the expression of matrix metalloproteinases (MMP-2 and MMP-9). Additionally, RA induces apoptosis through both extrinsic and intrinsic pathways. This comprehensive analysis positions RA as a promising therapeutic agent against metastatic colorectal cancer. The specific investigation on CT26 and HCT116 cells reinforces RA’s anti-proliferative effects, G0/G1 phase cell cycle arrest, and apoptosis induction.

In a study investigating the anti-Warburg effect of RA in gastric carcinoma, RA significantly inhibits MKN45 cell growth in a dose-dependent manner, with an IC50 value of 240.2 μM [41]. RA reduces glucose consumption, lactate generation, and downregulates HIF-1α expression, indicating an anti-Warburg effect. The study suggests that RA targets the glycolytic pathway and inflammatory pathways, such as IL-6/STAT3, through its impact on pro-inflammatory cytokines and microRNAs [41]. These findings position RA as a potential therapeutic agent for suppressing the Warburg effect in gastric carcinoma. Collectively, the studies underscore the promising anticancer properties of rosemary and its key compounds, *carnosic acid* (CA) and *rosmarinic acid* (RA), against various types of colon cancer, including colorectal and gastric carcinomas. CA and RA exhibit distinct yet complementary mechanisms of action. CA, through its inhibitory effects on cell growth, migration, and induction of apoptosis, proves effective against colorectal cancer, particularly when delivered via bovine serum albumin (BSA) nanoparticles. RA, on the other hand, demonstrates multifaceted therapeutic potential, inhibiting colorectal cancer cell proliferation, inducing apoptosis, and suppressing metastatic behavior by regulating epithelial-mesenchymal transition (EMT) and the Warburg effect. Furthermore, the rosemary extract, as explored by Pérez-Sánche et al. [39], significantly inhibits tumor growth and colony formation in colon cancer cells, showcasing its potential as a comprehensive anticancer agent. While each component presents unique characteristics, collectively, they provide a compelling foundation for further research into the development of rosemary-based formulations as promising therapeutic strategies against various cancers.

### 3.5. Prostate Cancer

Prostate cancer stands out as a pervasive malignancy exerting its impact on the prostate gland, solidifying its status as one of the most commonly diagnosed cancers in men on a global scale. In this section, we delve into the promising therapeutic dimensions of natural compounds, specifically focusing on rosemary extract, *carnosic acid*, and *rosmarinic acid*, within the intricate landscape of prostate cancer.

A comprehensive investigation by Jaglanian et al. [42] sheds light on the profound effects of rosemary extract on PC-3 prostate cancer cells. The study reveals a remarkable dose-dependent inhibition of cell proliferation, accentuated by a notable half maximal inhibition concentration (IC50) of 19.72 μg/mL [42]. Rosemary extract not only curtails cell migration but also orchestrates impactful changes in key signaling molecules, accompanied by elevated cleaved PARP levels, indicative of heightened apoptosis. These findings underscore the therapeutic potential of rosemary extract in thwarting prostate cancer progression, showcasing its superiority over conventional treatments [42].

Turning our attention to *carnosic acid*, its efficacy against prostate cancer becomes evident through studies highlighting its potent anticancer properties. Experiments on DU-145 and PC3 cell lines demonstrate significant growth inhibition and cytotoxic effects [43]. Impressively, even at concentrations as low as 6.25 μg/mL, marked decreases in cell viability are observed, with 80 μM inducing over 95% and 90% cell death in PC3 and DU-145 cells, respectively. *Carnosic acid* achieves apoptosis through intricate mechanisms, involving DNA fragmentation and modulation of key apoptotic proteins. The study underscores its versatility as a therapeutic agent by highlighting its ability to inhibit specific signaling pathways in the complex landscape of prostate cancer [43].

In the realm of rosemary compounds, *rosmarinic acid*, another compound from rosemary, demonstrated promising anticancer potential in studies on pancreatic cancer cell lines [43]. This compound induced cell cycle arrest and apoptosis, crucial processes in impeding cancer growth. The modulation of HDAC2 expression further *links rosmarinic acid* to the regulation of cancer progression [43]. While these findings highlight its potential as a therapeutic agent for pancreatic cancer, continued research is essential to comprehensively understand its efficacy and safety in this context. The determined IC50 value of *rosmarinic acid* in Hep-G2 human liver cancer cells, at 14 μM, adds valuable insights into its potential anticancer properties [44,45,46,47,48,49,50,51,52,53].

## 4. Theoretical Study

### 4.1. Compounds Preparation

In this theoretical study, the primary constituents of rosemary, namely, CA and RA, were investigated along with their anticancer activities against S100 and an inflammation inducing protein. The compounds, with CID 65126 and 5281792, respectively, were obtained from the PubChem database (http://pubchem.ncbi.nlm.nih.gov) (accessed on 4 Octobre 2023) [54]. Their 3D structures were prepared in Structure Data File (SDF) format and used for molecular docking studies. The ligands were optimized and minimized using the LigPrep module in the Maestro 12.8 program(Schrodinger 2021-2, New York City) [55], with the OPLS_2005 force field [56]. Hydrogen atoms were added, and salt and ionization were removed at pH (7 ± 2).

### 4.2. Sequence Alignment and Molecular Docking

The proteins under investigation encompassed the S100A8 homodimer, S100A9 homodimer, and the S100A8-S100A9 heterodimer complex. These proteins are typically expressed in immune cells under normal conditions but show abnormal expression in cancer cells. We accessed their crystal structures from the Protein Data Bank (PDB) with the respective IDs: 1MR8, 1IRJ, and 4GGF (Figure 5). To prepare the protein structures, we employed the protein preparation wizard (Maestro 12.8, Schrodinger 2021-2).

Our preparatory steps involved the selection of the active site as the focal point, removal of ligands and water molecules, and the application of default energy minimization settings, which included limiting root-mean-square deviation (RMSD) to 0.3 Å. Additionally, we added hydrogen atoms and generated protonation states using Epik within a pH range of 7 ± 2 [57]. The grid dimensions were chosen to encompass all ligand atoms with an additional 10 Å margin in every direction. Further refinement of the protein structure was accomplished using the OPLS_2005 force field. Docking poses underwent scoring via the extra precision (XP) glide score to predict binding affinity in Kcal/mol. The final and most favorable pose, characterized by the lowest binding energy, was selected. This selected pose was then transformed into both two-dimensional and three-dimensional representations, highlighting the intricate interactions between the ligand and active site residues [58].

### 4.3. Docking Studies

Molecular docking showed that both compounds could bind to the ligand binding domain of S100A8. This is the first study to investigate this possible link. To understand the molecular interaction between the drugs and S100A8, a series of molecular docking analyses were performed using the available three-dimensional structures (PDBID: 1IRJ, 1MR8, and 4GGF) with two antioxidant, anti-inflammatory, antidiabetic, and anticancer compounds: CA and RA. The ligands showed different binding intensities in the active sites with the protein target molecules, except for CA with the 1MR8 protein. The prediction of the binding free energy parameters was done to assess the optimal binding of the ligand with the protein of interest. The ligand and the target protein interactions were visualized with Maestro 12.8, Schrodinger2021-2. The interaction profiles included hydrogen bonds, polar, hydrophobic, pi-pi, pi-cation, and other contacts (Table 2).

S100A8 (1MR8): RA binds with the active site of Dimer S100A8 via three conventional hydrogen bonds, two with GLN69(A) and another with GLN69(B), a salt bridge with LYS77(A), which binds through the hydrophobic interaction with LEU72(A), ILE73(A), ILE76(A), VAL80(A), LEU72(B), ILE73(B), ILE76(B), and VAL80(B), with a docking score of −4.688 kcal/mol.

S100A9(1IRJ): The molecule RA in the docked structure have a good binding affinity with the S100A9 dimer, which is −5.446 kcal/mol and stabilizes the active site due to five conventional hydrogen bonds, two with ASP67(D), two others with GLU60(E), and one with LYS57(E); and a salt bridge with LYS57(B). In addition, residues MET63(D) and ALA70(D) are involved in the hydrophobic properties. The CA has a predicted bond free energy of −4850 kcal/mol with the dimer S100A9. In addition, there is a formation of three conventional hydrogen bonds with LYS57(G), GLU56(G). and LYS72(D); a salt bridge with LYS57(E); a hydrophobic interaction observed with residue MET63(D); and a polar contact with GLN35 (D).

S100A8–S100A9 (4GGF): The two molecules interact only with the K and S chains of S100A8–S100A9 for the CA, showing three conventional hydrogen bonds, two with ARG31(k) and one with ASP32(S); a salt bridge with LYS35(S); and two hydrophobic interactions with PHE55(K) and TYR30(K). Docking results are promising, with binding free energy estimates of −3.506 kcal/mol. RA reacts with the critical amino acid residues of the binding domain, enhancing the inhibitory capacity. The estimated very good binding free energy is predicted at −6720 kcal/mol. The conventional hydrogen bond-interacting residues of the drug binding site are LYS35(S) and ASP33(S). In addition, there was a salt bridge with ARG31(K). Stacking attractive pi−cation interactions with ARG31(k) and several hydrophobic contacts are seen in the docked complexes such as TYR30(K), PHE55(K), ALA65(K), and ILE22(S) (Figure 6).

The computational analysis reveals how the two inhibitors bind to the protein targets and what influences their binding strength. Moreover, the interactions also help to keep the binding structures stable.

## 5. Conclusions

Cancer cells often have mutated signaling molecules that make them grow out of control and avoid apoptosis, which are major challenges for effective treatment. To explore novel avenues for cancer prevention and treatment, plant-derived compounds have garnered attention, offering potential anticancer agents and fresh mechanistic insights. Among these bioactive substances, *Rosmarinus officinalis* L. compounds, especially the main polyphenolic compounds CA and RA, exhibit formidable antioxidant properties, a feature that has garnered substantial attention within the realm of cancer research. Indeed, oxidative stress is a pivotal contributor to the development of cancer. The antitumor effect of these secondary metabolites has been related to diverse mechanisms. In the case of CA, it has the capacity to induce cell death of cancer cells through the rise in ROS levels within the cells, the inhibition of protein kinase AKT, the activation of autophagy-related genes (ATG), and the disrupt mitochondrial membrane potential. Regarding RA, its antitumor actions encompass apoptosis induction through caspase activation, the inhibition of cell proliferation by interrupting cell cycle progression and epigenetic regulation, antioxidative stress-induced DNA damage, and interference with angiogenesis to curtail tumor growth. To understand the molecular interaction between rosemary compounds (CA and RA) and a protein that is involved in cancer and inflammation, S100A8, a series of molecular docking analyses were performed using the available three-dimensional structures (PDBID: 1IRJ, 1MR8, and 4GGF). The ligands showed different binding intensities in the active sites with the protein target molecules, except for CA with the 1MR8 protein.

## Figures and Tables

**Figure 1 plants-13-00089-f001:**
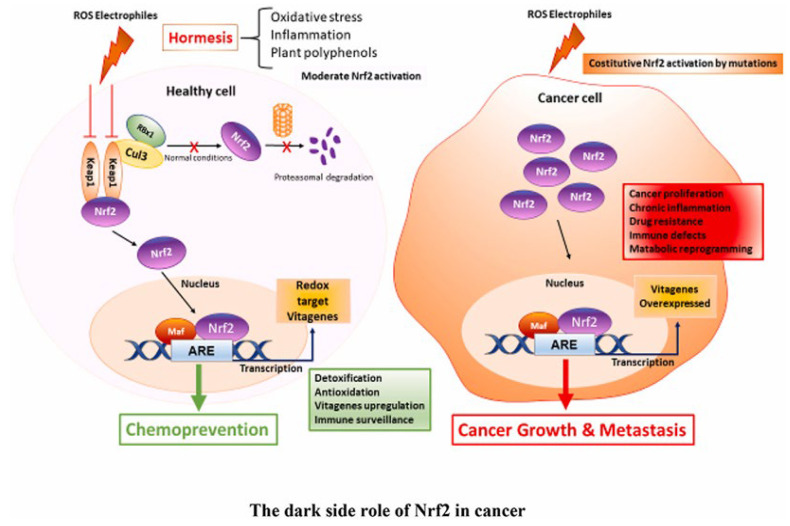
Mechanism of chemopreventive action of plant polyphenols. Edited from [20].

**Figure 2 plants-13-00089-f002:**
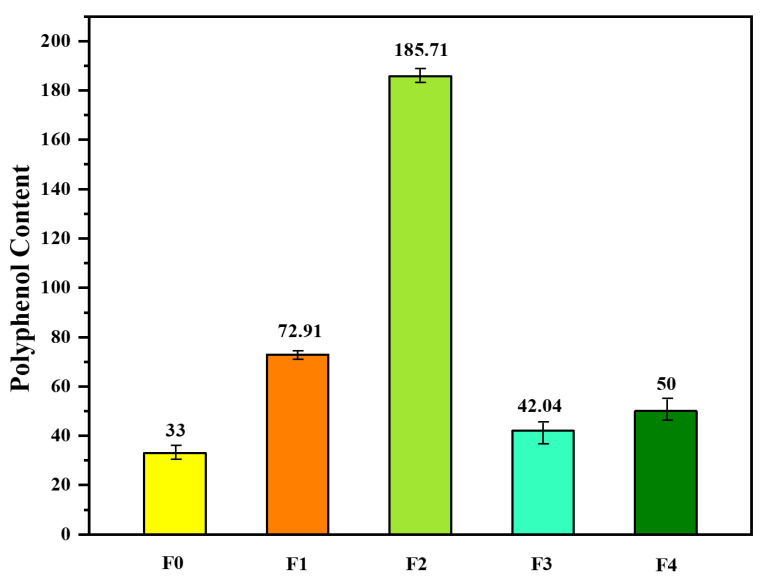
Polyphenol content expressed as gallic acid equivalents (mg GAE) per gram of *Rosmarinus officinalis* L. (F_0_ = methanol/water (80/20) extract; F_1–3_ are obtained from F_0_ by extraction using successively chloroform (F_1_), ethyl acetate (F_2_), and n-butanol (F_1_); F_4_ is the remaining final aqueous phase) [21].

**Figure 3 plants-13-00089-f003:**
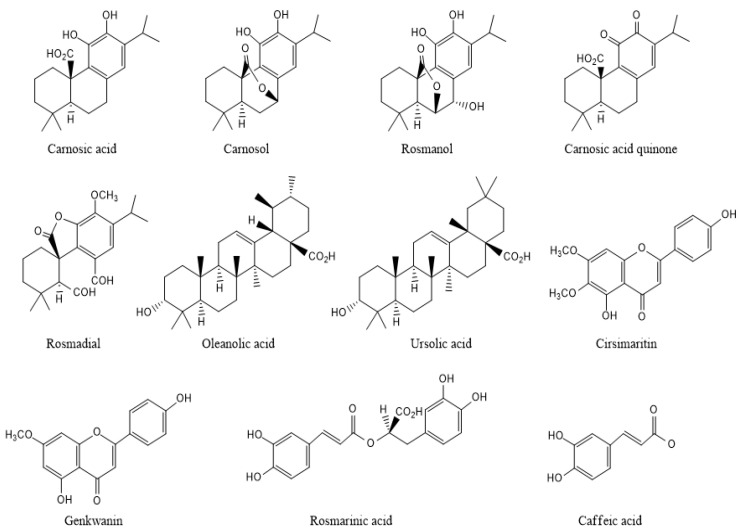
Chemical structures of varied bioactive compounds of rosemary.

**Figure 4 plants-13-00089-f004:**
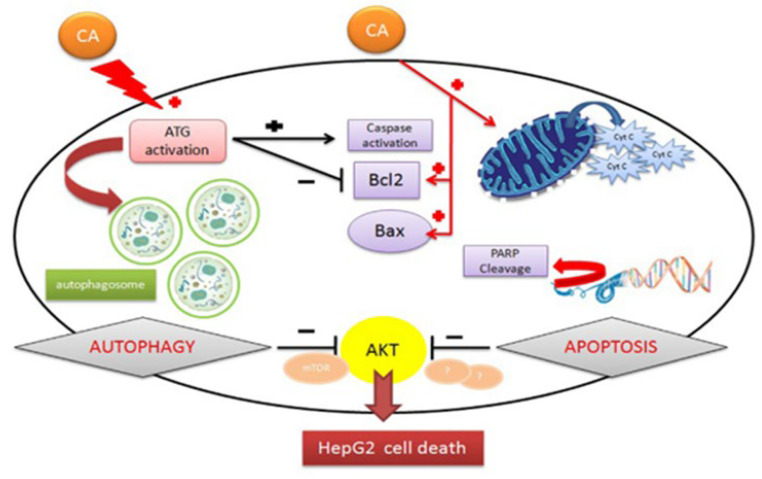
Anticancer mechanism of carnosic acid. Edited from [43].

**Figure 5 plants-13-00089-f005:**
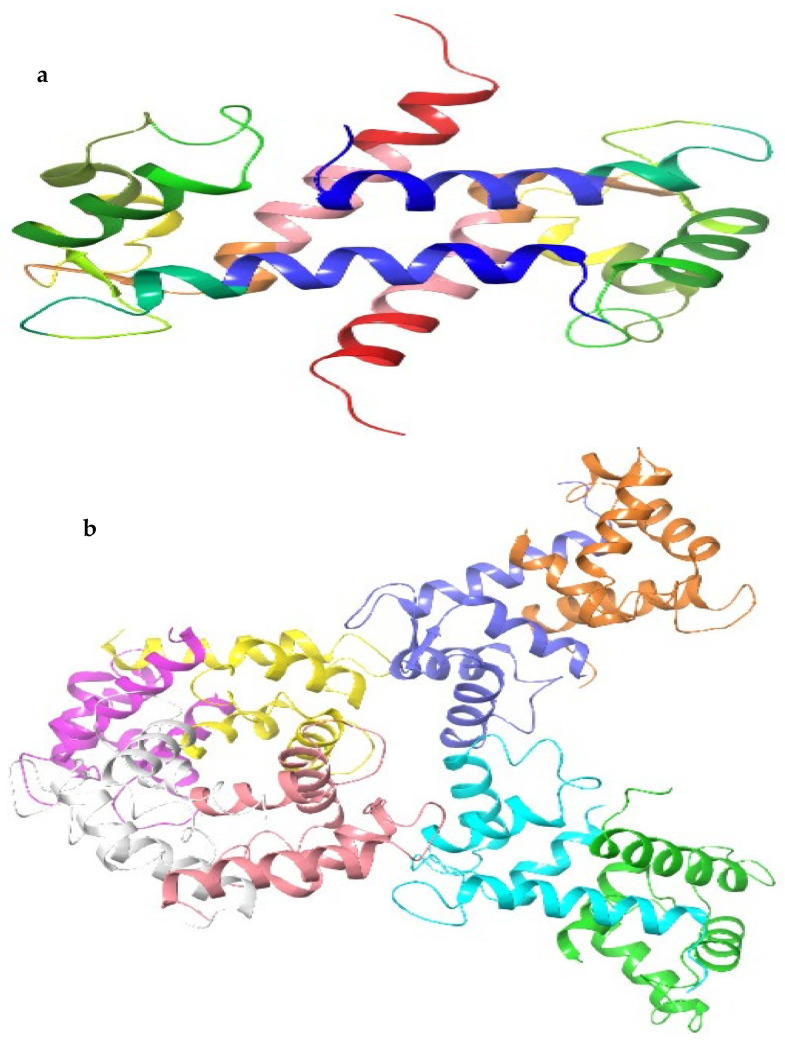
Structure of target proteins (**a**) S100A8 (1MR8), (**b**) S100A9 (1IRJ), and (**c**) S100A8−S100A9 (4GGF), involved in cancer and inflammation.

**Figure 6 plants-13-00089-f006:**
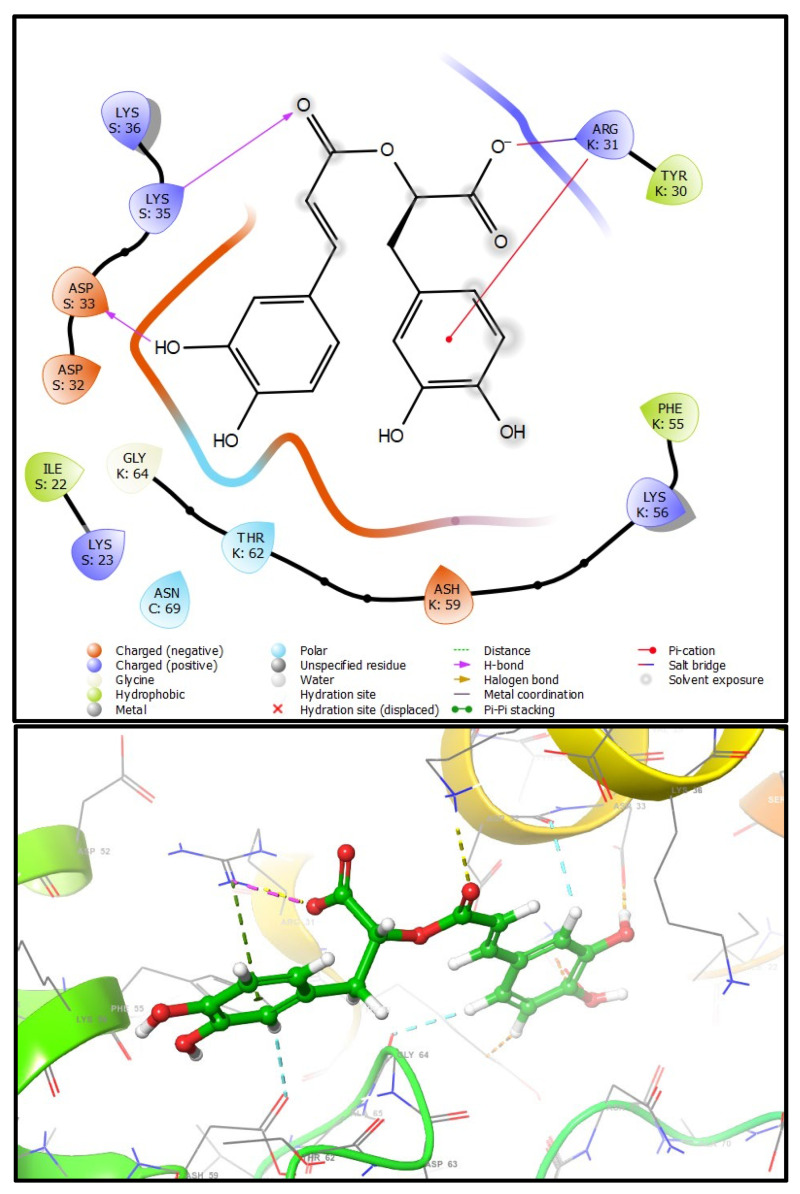
Intermolecular interactions (2D and 3D) between *rosmarinic acid* and S100A8-S100A9 heterodimer.

**Table 1 plants-13-00089-t001:** Antioxidant and antitumor properties of carnosic acid and rosmarinic acid derived from *Rosmarinus officinalis* L.

	Tested Compounds in IC50 (μΜ)			References
	Rosemary Extract	Carnosic Acid	Rosmarinic Acid	
**Radical-scavenging activity DPPH**	54.0 ± 1.4	33.1 ± 1.7	72.3 ± 3.3	[32]
**Oral cancer**	27.4–68.2 μM	7.5–20 μM	50–104.2 μM	[33,34,35]
**Kidney cancer**	-	20 μΜ	240.2 µM	[36,37]
**Pancreatic Cancer**	20–120 µM	6.02–54.15 µM	100 μΜ	[36,38]
**Colon cancer**	20–40 µg	8.29–27.43 mg/mL	240.2 μM	[39,40,41]
**Prostate cancer**	19.72 μg/mL	41.1 μM	14 μM	[42,43,44]

**Table 2 plants-13-00089-t002:** Binding free energy values for the docking of rosmarinic acid and carnosic acid with S100A8, S100A9, and S100A8–S100A9.

	Free Binding Energy	(kcal/mol)
	Rosmarinic Acid	Carnosic Acid
**S100A8**	−4.688	
**S100A9**	−5.446	−4.850
**S100A8–S100A9**	−6.720	−3.506

## Data Availability

Data will be made available on reasonable request.

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
