# Peer review of "Rosemary as a Potential Source of Natural Antioxidants and Anticancer Agents: A Molecular Docking Study"

_plants, 2023, doi:10.3390/plants13010089_

Round 1

Reviewer 1 Report

Comments and Suggestions for Authors

The topic is pretty relevant to the journal's readership and the author has also highlighted how this current research can impact the field. In general, I found the manuscript to be very well written, and all the sections were clear and concise and had a good flow between paragraphs. The title is clear and representative of the study. The abstract summarizes the work done concisely. The Introduction provides adequate background information for the reader and is well-framed and supported by data and citations. The conclusion adequately highlights the strengths of the study. However, I recommend minor revisions, and the authors need to address some queries.

1. Is figure 5 adopted from any sources? I do not see any citation for it.

2.  In the abstract, author mentioned "Among these compounds, rosemary (Rosmarinus officinalis L.) extract is a versatile candidate with antioxidant, anti-inflammatory, antidiabetic, and strong antitumor properties'. Did you want to say rosemary extract is a compound? 

3.  The author should enlist the constituents isolated from rosemary and should point out the structural characteristics of these components or their correlation with the anticancer mechanism. I look forward to the author's summary and critical insights on the available information, rather than a simple stack of data.

4. Is rosemary already important economically and socially in the countries in which it is used? Could it be? Justify and describe better the impact of the usage of this plant specie.

5. Reported clinical trials are limited. The authors need to be clear on this and give more examples of where products from this plant have successfully been used in a specific disease / clinical setting.

Comments on the Quality of English Language

Moderate.

Author Response

Responses to reviewers’ comments

Dear Editor/Reviewers

Thanks a lot for taking the time to revise and suggest these valuable comments, which will certainly improve the quality of our paper.

Accordingly, we addressed all the concerns and the detailed point-by-point responses are at the end of this letter. In this sense, all the changes made in the revised manuscript are highlighted.

 We hope all these modifications and revisions will be satisfactory to consider the present MS for publication in Plants journal.

Reviewer 1

Comment 1

Is figure 5 adopted from any sources? I do not see any citation for it.

Response:

This figure was replaced by Figure 3. Chemical structures of varied bioactive compounds of rosemary

Comment 2

In the abstract, author mentioned "Among these compounds, rosemary (Rosmarinus officinalis L.) extract is a versatile candidate with antioxidant, anti-inflammatory, antidiabetic, and strong antitumor properties'. Did you want to say rosemary extract is a compound? 

Response:

Abstract was corrected

Comment 3

The author should enlist the constituents isolated from rosemary and should point out the structural characteristics of these components or their correlation with the anticancer mechanism. I look forward to the author's summary and critical insights on the available information, rather than a simple stack of data.

Response:

The main constituents isolated from rosemary was enlisted

Comment 4

Is rosemary already important economically and socially in the countries in which it is used? Could it be? Justify and describe better the impact of the usage of this plant specie.

Response:

Nowadays, in the European Union, rosemary extracts are added to food products and beverages at levels of up to 400 mg/kg, , assigning them the label E392.

Comment 5

Reported clinical trials are limited. The authors need to be clear on this and give more examples of where products from this plant have successfully been used in a specific disease / clinical setting.

Response:

Dear Reviewer, thank you for your insightful comments and suggestions. We appreciate your attention to the reported clinical trials and agree that providing more examples of successful applications in specific diseases and clinical settings would enhance the comprehensiveness of our work. In response to your valuable feedback, we have incorporated additional data in our revised manuscript, focusing on the utilization of products from the plant in specific disease contexts. Notably, we have included comprehensive information on the plant's efficacy in the treatment of colon cancer and prostate cancer. These additions aim to address the limitations you pointed out and contribute to a more robust understanding of the potential clinical applications of the plant-derived products. We trust that these enhancements have strengthened the overall quality and relevance of our work. The added information is highlighted as in red text in the updated manuscript (please see the revised version)

Reviewer 2 Report

Comments and Suggestions for Authors

1)English correction should be done.

2) line 63: what type of biological effects?

3) line 48-53: reject this part of manuscript or add more information about polyphenols

4) line 92: lack of references

5) line 125: rosemarinic or rosmarinic?

6) why did you choose only carnosic and rosmarinic acids?

7) Add more references and data about CA and RA in plant.

Comments on the Quality of English Language

English is very difficult to understand that is why should be correct. Some information the Authors should add to have possibility to publish.

Author Response

Responses to reviewers’ comments

Dear Editor/Reviewers

Thanks a lot for taking the time to revise and suggest these valuable comments, which will certainly improve the quality of our paper.

Accordingly, we addressed all the concerns and the detailed point-by-point responses are at the end of this letter. In this sense, all the changes made in the revised manuscript are highlighted.

 We hope all these modifications and revisions will be satisfactory to consider the present MS for publication in Plants journal.

Reviewer 2

Comment 1

English correction should be done.

Response

The article has been revised by English professors.

Comment 2

line 63: what type of biological effects?

Response

Paragraph was deleted

Comment 3

 line 48-53: reject this part of manuscript or add more information about polyphenols

Response

Paragraph was corrected

Comment 4

line 92: lack of references

Response

We have fix these references, please see the revised version

Comment 5

 line 125: rosemarinic or rosmarinic?

Response

rosmarinic

Comment 6

 why did you choose only carnosic and rosmarinic acids?

Response

They were chosen for the following reasons:

- The principal bioactive components of the rosmary extracts are carnosic acid (CA, rosmarinic acid (RA) and carnosol (CAR)

- CA and RA, as major polyphenolic constituents in rosemary extract, exhibit formidable antioxidant properties.

- CA exhibits a diverse repertoire of benefits, including anti-inflammatory, antiproliferative, antitumorigenic, and neuroprotective properties. While RA, is shines with its antiviral, antibacterial, anti-inflammatory, and antioxidant effects

Comment 7

 Add more references and data about CA and RA in plant.

Response

We have add more references please see the revised version

Reviewer 3 Report

Comments and Suggestions for Authors

Latin names are written in italics.

To be corrected: - rosmarinic acid

In Figure 2 (Polyphenol content expressed as gallic acid equivalents (mg GAE) per gram of Rosmarinus officinalis.) the abbreviations F0-F4 are not explained, where the past values come from, a graph does not have a bibliographic index.

To create a table that centralizes the published information regarding the topic of the manuscript: e.g. extract prepared from ....with which solvent or RA or CA/action/mechanism/study (in vivo, in vitro)/ type of cancer/authors etc.

According to Fig. 2 follows Fig. 5 - Main phenolic compounds in Rosmarinus officinalis extract, rosmarinic and carnosic acids, as revealed by HPLC-DAD chromatogram profile. Please correct the numbers of the figures and add a bibliographic index.

From the sentence: "In this study, two of the rosemary extracts, CA and RA, were investigated along with their anticancer activities against S100, an inflammation inducing protein." it is understood that 2 rosemary extracts were studied. How were they prepared, in what solvent?

Please check the Bibliography and respect the rules of the journal.

Author Response

Responses to reviewers’ comments

Dear Editor/Reviewers

Thanks a lot for taking the time to revise and suggest these valuable comments, which will certainly improve the quality of our paper.

Accordingly, we addressed all the concerns and the detailed point-by-point responses are at the end of this letter. In this sense, all the changes made in the revised manuscript are highlighted.

 We hope all these modifications and revisions will be satisfactory to consider the present MS for publication in Plants journal.

Reviewer 3

Comment 1

Latin names are written in italics.

Response

Corrected, please see the revised version

Comment 2

To be corrected: - rosmarinic acid

Response

Corrected, please see the revised version

Comment 3

In Figure 2 (Polyphenol content expressed as gallic acid equivalents (mg GAE) per gram of Rosmarinus officinalis.) the abbreviations F0-F4 are not explained, where the past values come from, a graph does not have a bibliographic index.

Response

The below data has been added to Figure 2 :

F0 = Methanol/water (80/20) extract ; F1-3 are obtained from F0 by extraction using successively chloroform (F1), ethyl acetate (F2) and n-butanol (F1) ;  F4 is the remaining final aqueous phase.

Comment 4

To create a table that centralizes the published information regarding the topic of the manuscript: e.g. extract prepared from ....with which solvent or RA or CA/action/mechanism/study (in vivo, in vitro)/ type of cancer/authors etc.

Response

???

Comment 5

According to Fig. 2 follows Fig. 5 - Main phenolic compounds in Rosmarinus officinalis extract, rosmarinic and carnosic acids, as revealed by HPLC-DAD chromatogram profile. Please correct the numbers of the figures and add a bibliographic index.

Response

This figure was replaced by Figure 3. Chemical structures of varied bioactive compounds of rosemary

Comment 6

Line 371 : From the sentence: "In this study, two of the rosemary extracts, CA and RA, were investigated along with their anticancer activities against S100, an inflammation inducing protein." it is understood that 2 rosemary extracts were studied. How were they prepared, in what solvent?

Response

In this study, no separation of the two compounds has been carried out. However, we investigated their Docking studies separately according to the procedure described in the section 4 in view of determining their individual effectiveness against S100 and inflammation inducing protein.

 Comment 7

Please check the Bibliography and respect the rules of the journal.

Response

We have added the DOI, and followed the journal template

Round 2

Reviewer 2 Report

Comments and Suggestions for Authors

The manuscript can be published in Plants.

Reviewer 3 Report

Comments and Suggestions for Authors

The authors made the corrections and additions, and it seems that the manuscript could be published.